# Antimicrobial Resistance of *Escherichia coli* Isolates from Livestock and the Environment in Extensive Smallholder Livestock Production Systems in Ethiopia

**DOI:** 10.3390/antibiotics12050941

**Published:** 2023-05-22

**Authors:** Biruk Alemu Gemeda, Barbara Wieland, Gezahegn Alemayehu, Theodore J. D. Knight-Jones, Hiwot Desta Wodajo, Misgana Tefera, Adem Kumbe, Abebe Olani, Shubisa Abera, Kebede Amenu

**Affiliations:** 1Animal and Human Health Research Program, International Livestock Research Institute (ILRI), Addis Ababa P.O. Box 5689, Ethiopia; gezahegn.alemayehu@cgiar.org (G.A.); t.knight-jones@cgiar.org (T.J.D.K.-J.); h.desta@cgiar.org (H.D.W.); k.amenu@cgiar.org (K.A.); 2College of Veterinary Medicine and Agriculture, Addis Ababa University, Bishoftu P.O. Box 1176, Ethiopia; misganatefera3@gmail.com; 3Institute of Virology and Immunology, 3147 Mittelhaeusern, Switzerland; barbara.wieland@ivi.admin.ch; 4Department of Infectious Diseases and Pathobiology (DIP), Vetsuisse Faculty, University of Bern, 3012 Bern, Switzerland; 5Oromia Agricultural Research Institute, Yabello Pastoral and Dryland Agriculture Research Center, Yabello P.O. Box 85, Ethiopia; ademkumbe7@gmail.com; 6Animal Health Institute (AHI), Sebeta P.O. Box 04, Ethiopia; abebenaol@gmail.com (A.O.); shubisaabera12@gmail.com (S.A.)

**Keywords:** antimicrobial resistance, livestock, soil, *E. coli*, smallholders

## Abstract

The objective of this study was to characterize the distribution of antimicrobial resistance (AMR) of *Escherichia coli* (*E. coli*) isolated from livestock feces and soil in smallholder livestock systems. A cross-sectional study was carried out sampling 77 randomly selected households in four districts representing two agroecologies and production systems. *E. coli* was isolated and the susceptibility to 15 antimicrobials was assessed. Of 462 *E. coli* isolates tested, resistance to at least one antimicrobial was detected in 52% (43.7–60.8) of isolates from cattle fecal samples, 34% (95% CI, 26.2–41.8) from sheep samples, 58% (95% CI, 47.9–68.2) from goat samples and 53% (95% CI, 43.2–62.4) from soil samples. AMR patterns for *E. coli* from livestock and soil showed some similarities, with the highest prevalence of resistance detected against streptomycin (33%), followed by amoxycillin/clavulanate (23%) and tetracycline (8%). The odds of detecting *E. coli* resistance to ≥2 antimicrobials in livestock fecal samples were nearly three times (Odd Ratio—OR: 2.9; 95% CI, 1.72–5.17; *p* = 0.000) higher in lowland pastoral than in highland mixed crop–livestock production systems. These findings provide insights into the status of resistance in livestock and soil, and associated risk factors in low-resource settings in Ethiopia.

## 1. Introduction

Antimicrobial resistance (AMR) has been recognized as one of the most significant threats to the health of people and food-producing animals. The report from the World Health Organization (WHO) on AMR indicates that resistance of common bacteria has reached alarming levels in many parts of the world. For example, the resistance of *Escherichia coli* (*E. coli*) and *Klebsiella* spp. to last-resort third-generation cephalosporins and carbapenems antibiotics has reached up to 54% [1,2]. Some reports estimated that the economic loss due to AMR will increase dramatically, causing trillion-dollar losses by the mid-21st century [3]. In line with this, the 2030 Agenda for Sustainable Development Goals emphasized the need to address growing antimicrobial resistance [4,5].

AMR of human pathogens is inter-linked with AMR of bacteria associated with livestock, as well as the wider environment. Infections and resistance that originate in humans, animals, foods and farm environments will inevitably lead to the dissemination of infection with resistant bacteria and resistance genes in the wider environment [6,7]. This spreading of resistance may be facilitated by excreta coming into contact with soils as well as surface and ground water [8]. The human acquisition of AMR from domestic animals could occur by consuming contaminated animal-sourced foods, contaminated water, contact with domestic animals or contact with contaminated environments [9,10,11,12].

Cross-species transfer of resistant bacteria or resistance genetic elements from animals or the environment to humans has been reported [13,14,15]. The public health risks of a possible transfer of resistant zoonotic agents from animals to humans led to policy changes, such as a ban on the use of antibiotics as growth promoters in the European Union, and the introduction of AMR monitoring systems in livestock food systems in many countries [16]. Sweden was the first country to ban the use of antibiotics for growth promotion in food animal production in early 1986 [17]. Early insight about the risks of AMR was the main reason for the decision [18].

*E. coli* is one of the most widespread bacteria throughout the world. It is a normal commensal microbiota of the intestinal tract of animals and humans. However, not all *E. coli* strains are harmless, as some are able to cause diseases in humans as well as in mammals and birds [19,20]. For example, *E. coli* is a leading cause of extra-intestinal infections with strains colonizing the gastrointestinal tract of patients [21]. Animals are recognized as a reservoir for both human intestinal and extraintestinal pathogenic *E. coli* [22]. Enterohemorrhagic *E. coli* (EHEC), in particular EHEC O157:H7, a subtype of Shiga toxin producing *E. coli* (STEC), is of particular concern. AMR has been reported in *E. coli* from various animal species, the environment and in hospitalized patients from across the world, with many strains exhibiting multi-drug resistance (MDR) [21]. The rapid emergence of multi-drug-resistant pathogens has become one of the greatest concerns, as there are fewer, or even sometimes no, effective antimicrobial agents available for infections caused by these bacteria [23].

Given its widespread occurrence and capacity to assimilate resistances, *E. coli* has proved useful as a sentinel for monitoring antimicrobial drug resistance in fecal bacteria [24]. Studying AMR in *E. coli* is also important since the transmission of AMR through the environment is probably more important in *E. coli* than any other member of the microbiota. *E. coli* can remain viable in the environment (secondary habitat) for extended periods of time [25]. In addition, most AMR in *E. coli* is encoded on mobile genetic elements that are transferable between bacteria, thus enabling the rapid dissemination and maintenance of resistance genes between bacteria of different species (horizontal transfer of AMR genes) [10,26,27,28,29]. Therefore, AMR in *E. coli* is regarded as a major threat to public health [25].

The dynamics of AMR in developing countries are poorly understood, especially in rural community settings, due to a lack of data on the prevalence of AMR and molecular characteristics [21]. Although the surveillance capacity for AMR is minimal in most East African countries, and current data on AMR patterns of common pathogenic bacteria are sparse, high levels of AMR to commonly used antibiotics have been reported in this region [1,30]. AMR in Ethiopia is also increasing at an increasing rate [31,32,33,34]. While several separate studies have been conducted over the years on human patients, livestock, foods and the environment, with results indicating that there is a danger of losing worthy therapeutics due to the development of AMR by microorganisms, there are no robust national antimicrobial susceptibility data to show trends. A recent systematic review and meta-analysis revealed estimates of high AMR prevalence in bacteria from live animals, foods of animal origin, food handlers and the environment [35].

Attempts have been made to identify *E. coli* strains, particularly O157:H7, in meat and abattoir environments in Ethiopia and to understand its ecology and epidemiology, including sources of contamination, prevalence and antibiotic resistance profiles [36,37,38,39]. Yet, the extent to which livestock feces can serve as a source of resistant *E. coli* is poorly understood. However, this understanding is essential to develop effective strategies to reduce the emergence and spread of resistance, which is a global priority. This paper characterized the distribution of AMR of *E. coli* isolated from livestock feces and soil in a low-resource, extensive smallholder livestock production system.

## 2. Results

### 2.1. Occurrence of E. coli and E. coli O157:H7

One or more *E. coli* was isolated in 131 (86.2%) cattle fecal samples, 140 (77.8%) sheep samples, 89 (68.5%) goat samples and 51 (39.2%) soil samples.

The Biolog system (Omni Log ID) laboratory reader identified 42 (9.1%) of the 462 isolates as *E. coli* O157:H7. Table 1 presents the occurrence of *E. coli* O157:H7 by sample types and species of animals examined. The higher occurrence of *E. coli* O157:H7 was found among isolates from goats (20.2%), cattle (11.4%) and soil samples (6.8%). Conversely, only two isolates (1.4%) were characterized as *E. coli* O157:H7 from sheep samples.

### 2.2. Occurrence of Antimicrobial Resistance in E. coli

Of 462 *E. coli* isolates tested, resistance to at least one antimicrobial was found in 51.9% (43.4–60.3) of isolates from cattle, 33.6% (95% CI, 26.2–41.8) from sheep, 58.4% (95% CI, 47.9–68.2) from goats and in 52.9% (95% CI, 43.2–62.4) of soil samples, and a prevalence of 47.8 (95% CI, 43.3–52.4) across all livestock and soil samples combined was recorded.

The proportion of resistance to specific antibiotic classes among *E. coli* isolates from cattle, sheep, goat and soil was generally low (Table 2). Among 440 isolates, 23.2% showed resistance to amoxycillin/clavulanate, with 25.8% in cattle, 11.2% in sheep, 37.0% in goat and 24.8% in soil samples. Among 451 isolates tested for tetracycline, overall, 8% were resistant, with 3.9% resistance in cattle samples, 5.8% in sheep, 13.6% in goat and 11.3% in soil isolates. A higher level of resistance against streptomycin was found in the 449 isolates tested, with 33.2% of all samples testing positive, with 37.3% positive in cattle samples, 16.8% in sheep samples, 49.4% in goat samples and 36.4% in soil samples. A much smaller proportion of isolates were resistant against the other antibiotics tested (Table 2).

### 2.3. Multiple Antimicrobial Resistance

Out of 375 isolates tested for resistance against all 15 antibiotics, 205 (54.7%) of the isolates were susceptible to all of the antibiotics tested and 170 (45.3%) were resistant to at least one antibiotic. However, 100 (26.7%) showed multiple drug resistance (resistant to two or more antibiotics classes), with one sample resistant to 8 antibiotics classes. In total, 44 samples (11.8%) were resistant to ≥3 antibiotic classes, 18 (4.8%) were resistant to ≥4 and 11 (2.9%) were resistant to ≥5 (see Table 3). The most common co-resistant phenotype observed was resistance to amoxycillin/clavulanate and streptomycin (18.8%).

### 2.4. Antimicrobial Resistance among E. coli O157:H7 Isolates

Resistance among *E. coli* O157:H7 isolates was generally low regardless of the source of isolation. Among the 42 isolates, 5 (17.2%) were resistant to streptomycin and 3 (10.3%) were resistant to amoxycillin/clavulanate. Resistance to other antibiotics tested was rare, with resistance to chloramphenicol (3.5%) and cefotaxime (3.5%) detected at low levels. All isolates were susceptible to cefoxitin (second-generation cephalosporin).

Of all *E. coli* O157:H7 isolates, 19 (45.2%) showed MDR. The most common co-resistant phenotype observed was to amoxycillin/clavulanate and streptomycin (20.7%).

### 2.5. Risk Factors for the Occurrence of Antimicrobial Resistance Phenotypes

Univariable analyses showed that six variables were potentially associated (*p*-value < 0.25) with the occurrence of more than two or more antimicrobial resistance phenotypes in *E. coli* isolated from livestock, and seven variables for *E. coli* isolated from soil.

Table 4 shows risk factors associated with the occurrence of ≥2 antimicrobial resistance phenotypes among *E. coli* isolated from livestock using multivariable logistic regression analysis. The variables ‘manure management’ and ‘what do you do with dead animals?’ were both highly correlated with agroecology and not used in the model. Agroecology (pastoral system) and lack of access to professional animal health services were associated with *E. coli* resistance for ≥2 antibiotics in the models. The odds of detecting *E. coli* resistance to ≥2 antimicrobials in livestock fecal samples were nearly three times higher in lowland pastoral production systems than in highland mixed crop–livestock production systems (OR: 2.9; 95% CI, 1.72–5.17; *p* = 0.000).

Predictor variables for the occurrence of resistance to ≥2 antibiotics among *E. coli* isolate in soil samples are shown in Table 5. Manure management was retained in this model. Households that either used manure for fuel (incl. biogas) or sold manure for cash had a reduction of 85% (OR: 0.15; 95% CI, 0.03–0.75; *p* = 0.03) and had higher odds of having ≥2 antimicrobial resistance phenotypes in soil compared with those who left manure either on the farm or in the open air or discarded manure into the environment.

## 3. Discussion

We found a similar prevalence of resistant *E. coli* in livestock fecal samples and soil. Goat fecal samples had the highest prevalence of resistance (58%) to at least one antimicrobial in *E. coli,* followed by soil samples (53%). However, it is difficult to pinpoint the origin of the antimicrobial resistance observed. The lowest proportion of resistance was observed in isolates from sheep fecal samples. Despite the absence of drug stewardship in the study area, the resistance level for individual antibiotics tested was generally low, with a higher level of resistance against ‘older’ drugs such as streptomycin, followed by amoxycillin/clavulanate and tetracycline. This is expected, as resistance in *E. coli* mainly occurs against drugs that have been commonly used for farm animal treatments and/or prophylaxis for a long time [16,40,41,42]. Resistance against ‘newer’ drug classes (Cephalosporins, Quinolones, Chloramphenicol, Nitrofuran) was lower. This is in agreement with a recent systematic review and meta-analysis which noted that drug resistance in various samples, including animal-sourced foods, was against older drugs such as ampicillin, amoxicillin, streptomycin and tetracycline [35].

AMR occurrence in *E. coli* isolated from food-producing animals has been reported in different countries, but due to differences in sampling strategies, isolation methods and methods of AMR phenotype determination comparisons between studies is difficult. A study in Kenya showed that *E. coli* resistance to aminoglycosides, sulfonamides, tetracyclines, trimethoprim and penicillin was high in both humans and livestock, while resistance to cephalosporins and fluoroquinolones was low [43]. There is also evidence from community settings within countries in sub-Saharan Africa and in South Asia where *E. coli* resistances to ‘older’ antimicrobials was common, with 65% of isolates resistant to ampicillin, 67% to trimethoprim, 66% to trimethoprim/sulphamethoxazole, 56% to tetracycline and 43% resistant to streptomycin [21]. In a study in the USA, out of 746 *E. coli* isolates recovered from animal sources, 71.1% were resistant to tetracycline, 59% to streptomycin, 57.7% to sulfonamide and 34.1% to ampicillin [24].

Multi-drug-resistant pathogens have emerged worldwide [16]. In our study, the prevalence of MDR in *E. coli* was 26.7% and the most common co-resistant phenotype observed was to amoxycillin/clavulanate and streptomycin (18.8%). A relatively larger proportion of MDR *E. coli* isolates was recovered from animals in a US study [24]. They found that concurrent resistance to tetracycline and streptomycin was the most common co-resistance phenotype (30%), followed by resistance to tetracycline and sulfonamide (29%).

In our study, 42 (9%) *E. coli* isolates from livestock feces and soil were *E. coli* O157:H7, with higher proportions among isolates from goats and cattle. Hunduma (2018) also reported comparable prevalence of 4.7% *E. coli* O157 in both milk and feces samples in cows from a similar setting [44]. It is commonly cited that cattle are the primary reservoir of *E. coli* O157:H7 [45], with small ruminants also implicated [46,47,48]. The resistance level among *E. coli* O157:H7 isolates was generally low regardless of the source of isolation. Resistance to streptomycin (17%) and amoxycillin/clavulanate (10.3%) were the most common resistance profiles seen in these isolates. However, Hunduma found a higher level of *E. coli* O157 resistance to streptomycin (65%), tetracycline (59%) and Trimethoprim (24%) [44]. These drugs are still commonly administered in humans [49] and animals [41,42] in Ethiopia.

Bekele et al. reported *E. coli* O157:H7 isolates from raw meat in Addis Ababa that were resistant to different antibiotics including streptomycin (33%) and tetracycline (5%) [36]. With this study, we report for the first time AMR in *E. coli* O157:H7 isolated from soil, and thus confirm that soil contaminated via feces can act as a source of drug-resistant microbial pathogens including *E. coli* O157:H7. Greater attention should be paid to prevent *E. coli* O157:H7 contamination of the human food chains, given its health impact. Many studies have shown that the survival of *E. coli* O157:H7 in soil can lead to contamination of drinking water, fruits and fresh vegetables and constitutes a major public health threat [9,50,51,52,53,54]. Furthermore, *E. coli* O157:H7 can cause severe hemorrhagic colitis and hemolytic uremia in humans [55].

Our results also found higher odds for resistance in livestock from the lowland pastoral production system. This may be due to higher infection pressure and probability of recirculation of resistant isolates in the lowland agroecology and pastoral production system. It is possible that warm temperatures offer more potential for bacteria to multiply, with greater transference of antimicrobial resistance genes. Warmer temperatures are also associated with higher insect populations, which can play a role in disseminating resistant bacteria [56]. It could, however, also reflect the fact that improper use of antibiotics, mostly without a proper diagnosis, is more common in these production systems [42]. Such information is important to target AMR management practices.

Poor management, including the management and disposal of manure (i.e., leaving manure either on the farm, or in the open-air or discarding manure into the environment) was also strongly associated with detecting a higher level of *E. coli* resistance to more than two or more antimicrobial resistance phenotypes in soil samples. Similarly, Muloi et al. (2019) found that keeping manure inside the household compound was also significantly associated with AMR carriage in humans [43]. Animal manure has been implicated as a reservoir of AMR bacteria and AMR determinants [57,58]. Transmission of antibiotic-resistant *E. coli* and resistance genes may also occur through environments contaminated with feces, especially in developing countries [57,59].

Low levels of resistance are often overlooked, but can play an important role in the expansion of resistance [60]. In these extensive smallholder and pastoral settings, there is little to no testing of drug susceptibility during treatment of cases for both humans and animals. Hence, minimizing resistance is crucial. There is a need to maintain an overview of drug susceptibility through an AMR surveillance system that monitors resistance patterns and trends.

## 4. Materials and Methods

### 4.1. Study Area

The study was conducted in two agroecological zones and production systems: (i) a highland mixed crop–livestock production system (Menz Mama and Menz Gera district) and (ii) a pastoral system (Yabello and Eleweya districts). Details of the characteristics of the study area were published elsewhere [42]. Briefly, highland agroecology with a mixed crop–livestock system is typical for areas above 2200 m above sea level (masl) in which livestock husbandry depends on rain fed cropping. In lowland agroecology, pastoral livestock production is widespread, with the community mainly dependent on livestock and livestock products.

In recent publications, differences were reported between locations and production systems in terms of characteristics of antimicrobial usage including: (1) access to antimicrobials, (2) types of antimicrobials used and (3) when they were used. Livestock producers in mid/lowland pastoral systems appeared to use antibiotics more frequently than their counterparts in highland and lowland mixed crop–livestock systems [42].

### 4.2. Study Design and Sample Size Determination

A cross-sectional study was conducted with 77 households selected from extensive smallholder livestock systems in four districts. A total of 539 samples, which included 462 livestock fecal samples (cattle = 152, sheep = 180 and goats = 130) and 77 soil samples, were collected.

For fecal samples, the number of animals to be sampled in the study was estimated by the formula [61];
n = z2 × [Pexp (1 − Pexp)/d2](1)
where z = 1.96, Pexp (the expected prevalence) = 0.11 and d = 0.05 (the desired level of precision). Based on the result of a systematic review and meta-analysis, the overall pooled prevalence of *E. coli* expected was 15% in all samples [62]. The required sample size was, therefore, n = 231. To account for herd level clustering, the target sample size was adjusted using an intra-cluster correlation coefficient of 0.2 with an average of 6 animals sampled per herd. Accordingly, the design effect (D) of the study was calculated as 1.4 according to:D = 1 + (m − 1) × ρ(2)
where m was cluster size (i.e., 6), ρ was 0.2 and the calculated sample size was adjusted by multiplying by D. Therefore, the new sample size was 462 animals from 77 households.

Hence, 77 soil samples were collected from the homestead and barn areas of 77 households.

Household data were previously collected. In each household, details of household demographics, farm characteristics, manure management, feed types, animal health constraints, disease prevention, animal health services, antimicrobial use and animal product consumption were collected. Information on the selection of agroecological zones, districts and villages and random household selection was described in [42].

Because this study does not focus on the number of isolates per animal, we restricted the number of isolates to one or zero per animal.

### 4.3. Sample Collection and Pre-Enrichment Procedure

Fecal samples were taken from the rectum using a gloved hand and a sterile 50 milliliter (mL) capacity Falcon tube.

Soil samples were collected from either the homestead or the barn area of ruminants, from 2–5 cm beneath the surface. Approximately 10 g of soil, free of obvious fecal contamination, was collected into sterile vials. If the surface of the area was not flat, samples were collected from the lowest as well as middle and highest points, and mixed.

The fecal and soil samples were refrigerated and transported for laboratory analysis at either Yabello Pastoral and Dryland Agricultural Research Center (for samples collected from lowland pastoral areas) or the International Livestock Research Institute, Addis Ababa (for samples from highland agroecology) within 4–6 h of collection.

Immediately upon arrival at the lab, a sample suspension was prepared using 1 g of the sample in 9 mL of phosphate-buffered solution (5%). Samples were pre-enriched in buffered peptone water and incubated at 370 °C for 24 h.

### 4.4. Isolation and Identification of E. coli

A loop full of pre-enriched cultures was taken and inoculated on MacConkey agar and then incubated at 37 °C for 24 h. Typical colonies on MacConkey agar (pink, due to their ability to ferment lactose) were Gram-stained and observed for their staining and morphological characteristics, then transferred to eosin-methylene-blue (EMB) agar and incubated for 24 h at 37 °C. The colonies with metallic sheen on EMB agar, which is a typical characteristic of *E. coli*, were then considered as *E. coli*-positive and transferred to nutrient agar to be used for additional confirmatory biochemical tests (IMViC tests) and for further identification for Biolog tests.

Presumptive pure *E. coli* isolates were further analyzed using the Biolog system (OmniLog ID system, Hayward, CA, USA) following the standard procedures of the manufacturer. Briefly, the purified cultures of *E. coli* were inoculated on BUG (Biolog Universal Growth) agar medium; inoculums were prepared at a specified cell density using inoculating fluid A (IF A); the Biolog microplate GEN III was inoculated with the inoculums; the plate was incubated at 330 °C for 22 h into the Ominilog apparatus; the reaction pattern was entered; and results were obtained from the apparatus. The system also further identified *E. coli* O157:H7 serotype.

The purified cultures of *E. coli* were then stored in glycerol added to TSB broth at −200 °C for further biotyping and other studies.

### 4.5. Antimicrobial Susceptibility Test

The antimicrobial susceptibility test on the isolates was performed according to the National Committee for Clinical Laboratory Standards [63] using the Kibry-Bauer disk diffusion test method on Muller-Hinton agar (Oxoid CM0337 Basingstoke, England).

From each isolate, four to five confirmed colonies grown on nutrient agar were transferred to a test tube of 5 mL Tryptone Soya Broth (TSB) (Oxoid). The broth culture was incubated at 37 °C for 18 h until growth reached the 0.5 McFarland turbidity standard.

Mueller-Hinton agar plates were readied according to the manufacturer’s guidelines and held at room temperature for 30 min to allow drying. A sterile cotton swab was dipped into the suspension and then swabbed uniformly in three directions over the surface of Muller-Hinton agar plate (Oxoid Ltd, Hampshire, UK). After the plates dried, antibiotic disks were placed using an automatic Oxoid antimicrobial disk dispenser onto the inoculated plates and incubated at 37 °C for 24 h. After incubation for 24 h, the diameter of the zones of inhibition was measured and compared with zone size interpretation guidelines described by the Clinical Laboratory Standard Institute [64] for the family Enterobacteriaceae and determined as sensitive, intermediate or resistant.

The isolated *E. coli* were tested for sensitivity to the most commonly used antimicrobials. The zone of inhibition was interpreted based on the Performance Standards for Antimicrobial Susceptibility Testing; Sixteenth Informational Supplement, as detailed in Table 6.

### 4.6. Data Analysis

Overall and host- and sample-specific (livestock and soil) distribution of resistance phenotypes were determined by dividing the number of isolates with specific resistance phenotypes by the total number of isolates examined. Difference in proportions of samples in a group with resistance to one or more antibiotics was determined using Chi-squared tests and one-way analysis of variance (ANOVA; soil vs. different livestock groups). Bonferroni’s multiple-comparison test was performed post hoc for pairwise comparisons between groups, and *p*-values < 0.05 were considered significant.

Host- and sample-specific occurrence of *E. coli* O157:H7 were determined by dividing the number of *E. coli* O157:H7 isolated in each host by the total number of samples examined. The proportions of antimicrobial resistance were also determined by dividing the number of *E. coli* O157:H7 isolates with specific resistance phenotypes by the total number of isolates examined.

From the household data, eight variables were used as potential predictor variables for the occurrence of AMR. The variables were agroecology (highland mixed crop–livestock production system vs. pastoralist system); species mix (keep <3 livestock species vs. keep ≥3 livestock species); manure management (leave on farm or open air, discard into environment vs. use as fertilizer vs. use for fuel (incl biogas), sale for cash); isolation of sick animals (yes vs. no); allow mix of animals on treatment (yes vs. no); what do you do with dead animals (leave as it is vs. give to the dog vs. bury vs. human consumption); access to professional animal health services/treatments (yes vs. no); access to regular animal health services/vaccination and deworming (yes vs. no). A subset of the data that only included isolates tested for all antimicrobials (n = 288 for livestock and n = 87 for soil) was used for analysis of risk factors. To identify risk factors for having resistance to ≥2 antibiotics, odds ratios (ORs) were calculated using univariable logistic regression, followed by multivariable logistic regression.

Separate models were developed for livestock and soil, and a forward selection method with a significance level of 5% was used to select a suitable model. Models were adjusted for the cluster effect using robust standard error estimation for cluster sampling. When selecting important variables to the model, the Wald statistic associated with the variable was used instead of the likelihood ratio test (LRT) statistic [65].

Collinearity was assessed pair-wise via calculation of Spearman rank correlations between predictors. Variables with a *p*-value < 0.25 were considered for multivariable analysis, provided that there was no collinearity (r < |0.7|) between them. Potential confounders were considered in every model as variables that, if present, changed the coefficient for one or more significant variables by an amount that was important to report, taken as 10% [66]. All variables with a *p*-value ≤ 0.05 were retained in the final model. There were no biologically plausible interactions between the main effects expected and tested. Data were analyzed using Stata software version 16 (College Station, TX, USA).

## 5. Conclusions and Recommendations

These findings provide insights into the status of resistance in livestock and soil, and associated risk factors in low resource settings in Ethiopia. AMR patterns for *E. coli* from livestock and soil showed similarities. The resistance level for individual antibiotics tested was generally low, with the highest prevalence of resistance detected against streptomycin (33%), followed by amoxycillin/clavulanate (23%) and tetracycline (8%). The most common co-resistant phenotype observed was against amoxycillin/clavulanate and streptomycin. The findings showed that soil contaminated via feces can act as a source of drug-resistant microbial pathogens including *E. coli* O157:H7. The results also revealed that agroecology and production systems and manure management were strongly associated with occurrence of resistance. Additional molecular analysis of resistance genes is now planned to further investigate evidence of overlapping patterns of transferable resistance genes between livestock and soil. This would provide more definitive data on how resistance gene clusters have evolved and the context in which genes are maintained in the absence of known selection pressures.

In conclusion, we recommend that animal health and biosecurity practices, such as manure management, are radically improved, and that an integrated AMR surveillance system is established.

## Figures and Tables

**Table 1 antibiotics-12-00941-t001:** Occurrence of *E. coli* O157:H7 by sample types and species of animals examined.

Sample Type	Positives (%)	Odds Ratio	*p*-Value	CI for the Odds Ratio
Cattle feces	11.4 (7–18.1)	1.75	0.240	0.68–4.48
Sheep feces	1.4 (0.3–5.5)	0.19	0.045	0.04–0.96
Goat feces	20.2 (13.1–29.8)	3.4	0.009	1.36–8.68
Soil	6.8 (3.3–13.7)	Ref		

**Table 2 antibiotics-12-00941-t002:** Percentages of *E. coli* isolates resistant to different antibiotic classes classified by sample type (livestock spp. or soil).

Drug Class	Antibiotics	Host	n	Resistance Phenotypes in Different Host Types	
				% *	95% Conf. Interval
Penicillin	Amoxycillin/clavulanate (AML10)	Cattle	124	25.8 ^a^	18.9–34.2
		Sheep	134	11.2 ^b^	6.9–17.7
		Goat	81	37.0 ^a^	27.2–48
		Soil	101	24.8	17.3–34.1
		Total	440	23.2	19.4–27.4
Tetracyclines	Tetracycline (TE30)	Cattle	128	3.9 ^a^	1.6–9.1
		Sheep	138	5.8	2.9–11.2
		Goat	88	13.6 ^b^	7.9–22.5
		Soil	97	11.3	6.4–19.4
		Total	451	8	5.8–10.9
	Doxycycline (DO30)	Cattle	115	0.9	0.1–5.9
		Sheep	132	3	1.1–7.8
		Goat	67	0	-
		Soil	95	3.2	1.1–9.3
		Total	409	2	1–3.8
Fluoroquinolones	Ciprofloxacin (CIP5)	Cattle	127	2.4	0.7–7.1
		Sheep	137	4.4	1.9–9.4
		Goat	87	5.7	2.4–13.1
		Soil	99	5.1	2.1–11.6
		Total	450	4.2	2.7–6.5
Trimethoprim/Folate pathway inhibitors	Trimethoprim (W5)	Cattle	130	0 ^a^	-
		Sheep	136	2.2	0.7–6.6
		Goat	89	7.8 ^b^	3.8–15.6
		Soil	101	4.9	2.1–11.4
		Total	456	3.3	2–5.4
Sulfonamide	Sulfamethoxazole trimethoprim (SXT25)	Cattle	127	0 ^a^	-
		Sheep	137	3.6	1.5–8.5
		Goat	87	9.1 ^b^	4.6–17.4
		Soil	98	3.1	0.9–9.1
		Total	449	3.6	2.2–5.7
	Sulfonamide (S3_300)	Cattle	116	0.9 ^a^	0.1–5.8
		Sheep	133	1.5 ^ac^	0.4–5.8
		Goat	69	7.2	3–16.3
		Soil	93	8.6 ^b^	4.4–16.3
		Total	411	3.9	2.4–6.3
Aminoglycoside	Gentamicin (CN10)	Cattle	122	4.9	2.2–10.5
		Sheep	134	4.5 ^a^	2–9.6
		Goat	82	14.6 ^b^	8.5–24.1
		Soil	101	10	5.4–17.5
		Total	439	7.7	5.6–10.7
	Streptomycin (S25)	Cattle	126	37.3 ^a^	29.3–46.1
		Sheep	137	16.8 ^b^	11.4–24
		Goat	87	49.4 ^ac^	39.1–59.8
		Soil	99	36.4 ^ad^	27.5–46.3
		Total	449	33.2	28.9–37.7
Cephalosporins	Cefuroxime (CXM30)	Cattle	117	0.9	0.1–5.8
		Sheep	132	0	-
		Goat	74	4.1	13.1–11.9
		Soil	94	2.1	0.5–8.1
		Total	417	1.4	0.6–3.2
	Cefotaxime (CTX30)	Cattle	115	8.7	4.7–15.4
		Sheep	134	4.5	2.1–9.6
		Goat	69	4.3	1.4–12.7
		Soil	94	6.4	2.9–13.5
		Total	412	6.1	4.1–8.8
	Cefoxitin (FOX30)	Cattle	118	0.8	0.1–5.8
		Sheep	133	0.8	1.1–5.2
		Goat	70	0	
		Soil	92	1.1	0.2–7.3
		Total	413	0.7	0.2–2.2
Quinolones	Nalidixic acid (NA30)	Cattle	116	0	-
		Sheep	137	1.5	0.3–5.7
		Goat	72	1.4	0.2–9.2
		Soil	93	3.2	1–9.6
		Total	418	1.4	0.6–3.1
Chloramphenicol	Chloramphenicol (C30)	Cattle	124	1.6	0.4–6.2
		Sheep	137	2.2	0.7–6.6
		Goat	81	2.5	0.6–9.4
		Soil	93	2.1	0.5–8.2
		Total	435	2.1	1.1–3.9
Nitrofuran	Nitrofurantoin (F300)	Cattle	123	0	-
		Sheep	132	0.8	0.1–5.2
		Goat	83	2.4	0.6–9.2
		Soil	101	1.9	0.4–7.6
		Total	425	1.1	0.5–2.7

Each superscript letter (*) denotes a resistance phenotype percentage in different host types: the same letter or no letter means the percentages do not differ significantly from each other at the 0.05 level (ANOVA).

**Table 3 antibiotics-12-00941-t003:** Multiple antimicrobial resistance of *E. coli* isolated from livestock and soil.

Number of Antimicrobial Resistances	Predominant Resistance Profile Composition *	No. of Isolates (%)
Zero	-	205 (54.7)
One	S; Aml; Ctx; Te; Cn	170 (45.3)
Two	AmlS; SCtx; CipS; CnS;	56 (14.9)
Three	AmlCnS; TeSS3; AmlTeS; AmlSF; AmlCipS	26 (6.9)
Four	AmlCnSCxm; AmlCnTES	7 (1.9)
Five	WamlTeSxtS; AmlCnSCDo	7 (1.9)
Six	AmlCnSFCS3; AmlTeSxtCNaDo	3 (0.8)
Eight	WamlCnTeCipSxtSS3	1 (0.3)

* Amoxycillin/clavulanate (Aml); Cefotaxime (Ctx); Cefoxitin (F); Cefuroxime (Cxm); Chloramphenicol (C); Ciprofloxacin (Cip); Doxycycline (Do); Gentamicin (Cn); Nalidixic acid (Na); Streptomycin (S); Sulfamethoxazole trimethoprim (Sxt); Sulfonamide (S3); Tetracycline (Te); Trimethoprim (W).

**Table 4 antibiotics-12-00941-t004:** Potential predictor variables for the occurrence of ≥2 resistance phenotypes in livestock using multivariable logistic regression analysis.

Predictor	Level/Category	% AMR (95% CI)	Univariable Analysis			Multivariable Analysis		
			OR	95% CI	*p*-Value	OR	95% CI	*p*-Value
Agroecology	Highland mixed crop–livestock production system (n = 158)	7 (3.8–12.2)	Ref					
	Pastoralist system (n = 130)	43.8 (35.5–52.5)	3.2	2.3–4.6	0.000	2.98	1.72–5.17	0.000
Species mix	Keep <3 species (n = 39)	41 (26.8–56.9)	Ref					
	Keep ≥3 species (n = 244)	20.9 (16.2–26.5)	0.38	0.18–0.77	0.006	1.35	0.56–3.22	0.498
Manure mgt.	Leave on farm; Open air; Discard into environment (n = 125)	44.8 (36.3–53.6)	Ref					
	Used as fertilizer (n = 32)	15.6 (6.6–32.5)	0.23	0.09–0.68	0.004			
	Used for fuel (incl. biogas); Sold for cash (fuel) (n = 126)	4.8 (2.1–10.2)	0.06	0.04–0.20	0.000			
Isolation of sick animals	Yes (n = 150)	11.3 (7.1–17.5)	Ref					
	No (n = 133)	37.5 (29.7–46.1)	4.7	2.5–8.7	0.000	1.24	0.39–3.88	0.713
Allow mix of animals on treatment	Yes (n = 95)	36.8 (27.7–46.9)	Ref					
	No (n = 188)	17 (12.3–23.1)	0.35	0.2–0.62	0.000	0.62	0.32–1.17	0.143
What do you do with dead animals?	Leave as it is (n = 4)	75 (23.6–96.6)	22.2	1.73–59.1	0.010			
	Give to the dog (n = 112)	5.3 (2.4–11.4)	0.41	0.15–1.61	0.245			
	Bury (n = 42)	11.9 (5–25.6)	Ref					
	Human consumption (n = 125)	42.4 (34–51.2)	4.84	1.81–12.9	0.002			
Access to professional animal health services/treatments	Yes (n = 245)	22.8 (18.1–28.5)	Ref					
	No (n = 38)	28.9 (16.8–45.2)	1.4	0.64–2.94	0.3			
Access to regular animal health services (vaccination and deworming)	Yes (n = 256)	23.8 (18.9–29.4)	Ref					
	No (n = 27)	22.2 (10.3–41.5)	0.91	0.35–2.36	0.537			

**Table 5 antibiotics-12-00941-t005:** Potential predictor variables for the occurrence of ≥2 resistance phenotypes in soil using multivariable logistic regression analysis.

Predictor	Level/Category	% AMR (95% CI)	Bivariate Analysis			Multivariable Analysis		
			OR	95% CI	*p*-Value	OR	95% CI	*p*-Value
Agroecology	Highland mixed crop–livestock production system (n = 58)	24.1 (14.7–36.9)	Ref					
	Pastoralist system (n = 29)	62.1 (43.3–77.8)	2.3	1.4–3.6	0.001			
Species mix	Keep <3 species (n = 9)	55.5 (24.7–82.5)	Ref					
	Keep ≥3 species (n = 77)	35.1 (25.1–46.4)	0.43	0.11–1.74	0.239			
Manure mgt.	Leave on farm; Open air; Discard into environment (n = 27)	66.7 (47–81.8)	Ref					
	Used as fertilizer (n = 9)	44.4 (17.4–75.2)	0.4	0.08–1.86	0.243	0.47	0.12–1.885	0.291
	Used for fuel (incl. biogas); Sold for cash (fuel) (n = 49)	20.4 (11.2–34.2)	0.13	0.04–0.36	0.000	0.15	0.03–0.75	0.03
Isolation of sick animals	Yes (n = 52)	25 (14.9–38.6)	Ref					
	No (n = 34)	55.8 (38.9–71.6)	3.8	1.5–9.56	0.004	0.98	0.25–3.91	0.986
Allow mix of animals on treatment	Yes (n = 23)	47.8 (28.5–67.7)	Ref					
	No (n = 63)	33.3 (22.7–45.9)	0.54	0.21–1.44	0.221			
What do you do with dead animals?	Leave as it is (n = 1)	(no observation)	-					
	Give to the dog (n = 47)	25.5 (14.9–40)	1.54	0.29–8.17	0.61			
	Bury (n = 11)	18.2 (4.5–51.3)	Ref					
	Human consumption (n = 27)	66.7 (47–81.8)	9	1.59–50.7	0.013			
Access to professional animal health services	Yes (n = 75)	32 (22.3–43.4)	Ref					
	No (n = 11)	72.7 (40.9–91.1)	5.6	1.37–23.2	0.016	1.6	0.28–8.85	0.59
Access to regular animal health services (vaccination and deworming)	Yes (n = 76)	35.5 (25.4–47)	Ref					
	No (n = 10)	50 (22.1–77.8)	1.81	0.48–6.83	0.378			

**Table 6 antibiotics-12-00941-t006:** Zone interpretive chart for antimicrobials (inhibition zone diameter in mm).

Antimicrobial Agent	Disc Content (µg)	Resistant (≤)	Intermediate	Susceptible (≥)
Trimethoprim (W5)	5	10	11–15	16
Amoxycillin/clavulanate (AML10)	10	13	14–17	18
Gentamicin (CN10)	10	12	13–14	15
Tetracycline (TE30)	30	11	12–14	15
Ciprofloxacin (CIP5)	5	15	16–20	21
Sulfamethoxazole trimethoprim (SXT25)	25	10	11–15	16
Streptomycin (S25)	25	11	13–14	15
Cefuroxime (CXM30)	30	14	15–17	18
Nalidixic acid (NA30)	30	13	14–18	19
Chloramphenicol (C30)	30	12	13–17	18
Cefotaxime (CTX30)	30	22	23–25	26
Cefoxitin (FOX30)	30	14	15–17	18
Doxycycline (DO30)	30	10	11–13	14
Sulfonamide (S3_300)	300	12	13–16	17
Nitrofurantoin (F300)	300	14	15–16	17

## Data Availability

The datasets generated for this study are available on request to the corresponding author.

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
