# Peer review of "Antimicrobial Resistance of Escherichia coli Isolates from Livestock and the Environment in Extensive Smallholder Livestock Production Systems in Ethiopia"

_antibiotics, 2023, doi:10.3390/antibiotics12050941_

Round 1

Reviewer 1 Report

The manuscript titled “Antimicrobial Resistance of Escherichia coli Isolates from Livestock and the Environment in Extensive Smallholder Livestock Systems in Ethiopia” is interesting and can be accepted for publication after these major changes.

1. 77 smallholder households. Is this term “smallholder households” appropriate?

2. Overall the study lack the information on the genetic mechanisms of resistance as The study did not assess the genetic mechanisms of resistance, which could provide insight into the development and spread of AMR in the region.

3. The authors have mentioned “These findings provide insights into the emergence of resistance in livestock and soil, and associated risk factors in low resource settings in Ethiopia. How can the authors state its emergence of resistance? Has the resistance increased compared to previous studies of similar kind from the region?

4. Table 3 is not clear. Please check “Predominant resistance profile composition”

5. Please carefully check the whole manuscript as there are grammatical mistakes at certain places.

Please carefully check the whole manuscript as there are grammatical mistakes at certain places.

Author Response

Reviewer 1

The manuscript titled “Antimicrobial Resistance of Escherichia coli Isolates from Livestock and the Environment in Extensive Smallholder Livestock Systems in Ethiopia” is interesting and can be accepted for publication after these major changes.

  1. 77 smallholder households. Is this term “smallholder households” appropriate?

Response:

Yes, the term “smallholder households” might be confusing and not appropriate, we have rephrased it as “77 households selected from extensive smallholder livestock systems in four districts” in line 307-306.

  1. Overall the study lack the information on the genetic mechanisms of resistance as The study did not assess the genetic mechanisms of resistance, which could provide insight into the development and spread of AMR in the region.

Response:

Yes, we didn’t include analysis of resistance genes in this manuscript, but we plan to conduct molecular analysis of resistance gene. Therefore, we have indicated this in line 435-439 of the manuscript as below.

“Additional molecular analysis of resistance genes is now planned to further investigate evidence of overlapping patterns of transferable resistance genes between livestock and soil. This would provide more definitive data on how resistance gene clusters have evolved and the context in which genes are maintained in the absence of known selection pressures.”

  1. The authors have mentioned “These findings provide insights into the emergence of resistance in livestock and soil, and associated risk factors in low resource settings in Ethiopia. How can the authors state its emergence of resistance? Has the resistance increased compared to previous studies of similar kind from the region?

Response

Point taken. By doing only cross-sectional study and without showing the trend of AMR rate, we cannot claim the emergence of resistance in the study area, therefore we have now revised the conclusion and corrected the statement as “These findings provide insights into the status of resistance in livestock and soil, and associated risk factors in low resource settings in Ethiopia.” Please see revisions of the conclusion in truck changes.

  1. Table 3 is not clear. Please check “Predominant resistance profile composition”

Response

To make it clearer, we have now added footnotes to the table.

*Amoxycillin/clavulanate (Aml); Cefotaxime (Ctx); Cefoxitin (F); Cefuroxime (Cxm); Chloramphenicol (C); Ciprofloxacin (Cip); Doxycycline (Do); Gentamicin (Cn); Nalidixic acid (Na); Streptomycin (S); Sulfamethoxazole trimethoprim (Sxt); Sulfonamide (S3); Tetracycline (Te); Trimethoprim (W)

  1. Please carefully check the whole manuscript as there are grammatical mistakes at certain places.

Response

We have now reviewed the manuscript carefully and tried to correct grammatical errors. Please see the truck changes.

Reviewer 2 Report

The authors have decided to analyse the % of E.coli O157:H7 among isolated strains. However, later on they only show a collective data on antimicrobial resistance of all isolated E. coli. The reviewers wonders if the resistance of E. coli O157:H7 could also be shown separately to compare it with general data. This would increase the scientific value of the paper. Antibiotic therapy is usually not recommended to treat the infection in patients. However, this still happens due to lack of identification and also, this strain is still under selective pressure in the environment. 

My additional question is have the authors checked for the presence of plasmids or genes in bacteria to confirm the presence of the resistant genes?

The English language requires a spell check, there are some "a" and "the" missing in a few places.

Also, re-read the introduction and discussion as there are some words and phrases that could be ommited or rephrased to sound better i.e. line 50-51 having contact with uncooked meat or poultry, contact with domestic animals or contact with contaminated environments.

Author Response

Reviewer 2

The authors have decided to analyse the % of E.coli O157:H7 among isolated strains. However, later on they only show a collective data on antimicrobial resistance of all isolated E. coli. The reviewers wonders if the resistance of E. coli O157:H7 could also be shown separately to compare it with general data. This would increase the scientific value of the paper. Antibiotic therapy is usually not recommended to treat the infection in patients. However, this still happens due to lack of identification and also, this strain is still under selective pressure in the environment.

Response:

We appreciate the concern, but indeed the AMR level in E. coli O157:H7 was given emphasis and separately described in line 149-156 of page 5-6.

My additional question is have the authors checked for the presence of plasmids or genes in bacteria to confirm the presence of the resistant genes?

Response:

Not yet, we plan to conduct molecular analysis of resistance genes. Therefore, we have indicated this in line 435-439 of our manuscript as below.

“Additional molecular analysis of resistance genes is now planned to further investigate evidence of overlapping patterns of transferable resistance genes between livestock and soil. This would provide more definitive data on how resistance gene clusters have evolved and the context in which genes are maintained in the absence of known selection pressures.”

Comments on the Quality of English Language

The English language requires a spell check, there are some "a" and "the" missing in a few places.

Response:

Point taken, we reviewed the manuscript carefully and tried to correct English language errors.

Also, re-read the introduction and discussion as there are some words and phrases that could be ommited or rephrased to sound better i.e. line 50-51 having contact with uncooked meat or poultry, contact with domestic animals or contact with contaminated environments.

Response:

Corrected in line 48-50 as “The human acquisition of AMR from domestic animals could occur by consuming contaminated animal sourced foods, contaminated water, contact with domestic animals or contact with contaminated environments”.

Round 2

Reviewer 1 Report

The authors have made improvements to the manuscript based on the suggestions provided.

The English is fine, with only minor checks required.